# GATE-GUIDED AND SUBGRAPH-AWARE BILATERAL FUSION FOR MOLECULAR PROPERTY PREDICTION

## ABSTRACT

Predicting molecular properties is crucial in scientific research and industry applications. Molecules are often modeled as graphs where atoms and chemical bonds are represented as nodes and edges, respectively, and Graph Neural Networks (GNNs) have been commonly utilized to predict atom-related properties, such as reactivity and solubility. However, some properties, such as efficacy, and metabolic properties, are closely related to functional groups (subgraphs), which cannot be solely determined by individual atoms. In this paper, we introduce the Gate-guided and Subgraph-aware Bilateral Fusion (GSBF) model for molecular property prediction. GSBF overcomes the limitations of prior atom-wise and subgraph-wise models by integrating both types of information into two distinct branches within the model. We provide a gate-guided mechanism to control the utilization of two branches. Considering existing atom-wise GNNs cannot properly extract invariant subgraph features, we propose a decomposition-polymerization GNN architecture for the subgraph-wise branch. Furthermore, we propose cooperative node-level and graph-level self-supervised learning strategies for GSBF to improve its generalization. Our method offers a more comprehensive way to learn representations for molecular property prediction. Extensive experiments have demonstrated the effectiveness of our method.

## 1 INTRODUCTION

Molecular properties prediction plays a fundamental role in many tasks like drug and material discovery (Feinberg et al., 2018). Previous methods typically model molecules as graphs, where atoms and chemical bonds are modeled as nodes and edges, respectively. Therefore, Graph Neural Networks (GNNs) (Hamilton et al., 2017) have been widely applied to predict specific properties associated with atoms, such as solubility and reactivity (Zhang et al., 2021; Hu et al., 2019; Yang et al., 2022). For simplicity, we refer to these models as atom-wise models, as the final molecular representation is the average of all atoms.

However, not all molecular properties are determined by individual atoms, and some chemical properties are closely related to functional groups (subgraphs) (Yang et al., 2022; Kong et al., 2022). For example, a molecule's efficacy, and metabolic properties are often determined by the functional groups within it. Therefore, many methods propose subgraph-wise models (Jiang et al., 2023; Yang et al., 2022), where the final molecular representation is the average of all subgraphs. The method proposed in (Yang et al., 2022) decomposes a given molecule into pieces of subgraphs and learns their representations independently, and then aggregates them to form the final molecular representation. By paying more attention to functional groups, this method ignores the influence of individual atoms[1], which is harmful to predicting properties related to atoms. More commonly, different properties are sensitive to atoms or functional groups but often determined by both of them at the same time. As shown in Fig. 1, different properties are determined by atoms and subgraphs differently. To further verify our point, we do experiments on the BBBP and Tox21 datasets and visualize the performance on their test sets in Fig. 1 (b) and (c), respectively. It is shown that BBBP is more sensitive to the subgraph-wise branch while Toxcast is more sensitive to the atom-wise branch.

---

[1]The representation of each subgraph merges the atom-wise representation by the attention mechanism, but it does not model atom-representation relationships explicitly.

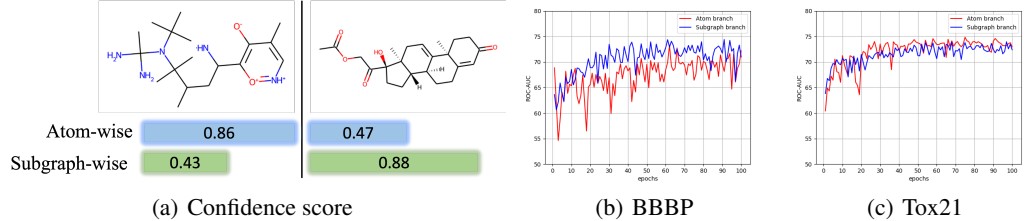

(a) Confidence score      (b) BBBP      (c) Tox21

Figure 1: (a) We select two molecules from BBBP (left) and Tox21 (right) and calculate the confidence score of the two branches, respectively. (b) Testing curves on the BBBP dataset. (c) Testing curves on the Tox21 dataset. This illustrative experiment indicates that different properties have different correlations with atom-wise and subgraph-wise information.

Therefore, both atom-wise and subgraph-wise models have inadequacies and cannot accurately predict current molecular properties independently. To address this dilemma, we propose our Gate-guided and Subgraph-aware Bilateral Fusion (GSBF), which incorporates the atom-wise and subgraph-wise information. We introduce a Gate-guided module that allows for the dynamic selection of atomic-level or subgraph-level information, catering to the specific requirements of molecular properties. This mechanism enhances adaptability in information processing for molecular properties prediction. For the subgraph-wise branch, we propose a Decomposition-Polymerization GNN architecture where connections among subgraphs are broken in the lower decomposition GNN layers and each subgraph is viewed as a single node in the higher polymerization layers. The decomposition layers serve as a separate GNN module that embeds each subgraph into an embedding, and the relations among subgraphs are modeled by polymerization layers. In this way, the second branch takes subgraphs as basic token units and aggregates them as the final representation. Similar to the atom-wise branch, an independent classifier is added to predict the property. Finally, we incorporate the outputs of the two branches as the prediction score.

In addition, we propose a corresponding self-supervised learning method to jointly pre-train the two branches of our GSBF. Existing self-supervised molecular learning methods mainly design atom-wise perturbation invariance or reconstruction tasks, e.g. predicting randomly masked atoms, but they cannot fully capture subgraph-wise information and the relations among substructures. To this end, we propose a Masked Subgraph-Token Modeling (MSTM) strategy for the subgraph-wise branch. MSTM first tokenizes a given molecule into pieces and forms a subgraph-token dictionary. Compared with atom tokens, such subgraphs correspond to different functional groups, thus their semantics are more stable and consistent. MSTM decomposes each molecule into subgraph tokens, masks a portion of them, and learns the molecular representation by taking the prediction of masked token indexes in the dictionary as the self-supervised task. For the atom-wise branch, we simply employ the masked atom prediction task. Although the atom-wise branch and the subgraph-wise branch aim to extract molecular features from different levels, the global graph representations for the same molecule should be consistent. To build the synergistic interaction between the two branches for joint pre-training, we perform contrastive learning to maximize the average invariance of the two branches. Experimental results show the effectiveness of our method.

Our contributions can be summarized as:

1. We propose a bilateral fusion model to encode the characteristics of both atoms and subgraphs with two branches adaptively. For the subgraph branch, we propose a novel decomposition-polymerization architecture to embed each subgraph token independently with decomposition layers and polymerize subgraph tokens into the final representation with polymerization layers. In addition, we propose a gate mechanism to adjust the fusion rate of the two branches.

2. We propose a cooperative node-level and graph-level self-supervised learning method to jointly train the two branches of our bilateral model. For the subgraph branch, we propose MSTM, a novel self-supervised molecular learning strategy, which uses the auto-discovered subgraphs as tokens and predicts the dictionary indexes of masked tokens. The subgraph tokens are more stable in function and have more consistent semantics. In this

way, masked subgraph modeling can be performed in a principled manner. At the global graph level, we perform a contrastive learning strategy that imposes the interaction of the two branches with the consistency constraint.

3. We provide extensive empirical evaluations to show that the learned representation by our bilateral model and our self-supervised learning method has a stronger generalization ability in various functional group-related molecular property prediction tasks.

## 2 RELATED WORK

**Molecular Property Prediction**   The prediction of molecular properties is an important research topic in the fields of chemistry, materials science, pharmacy, biology, physics, etc (Wang & Hou, 2011). Since it is time-consuming and labor-intensive to measure properties via traditional wet experiments, many recent works focus on designing end-to-end machine learning methods to directly predict properties. These works can be divided into two categories: SMILES string-based methods (Butler et al., 2018; Dong et al., 2018) and graph-based methods (Gilmer et al., 2017; Yang et al., 2019; Lu et al., 2019; Gasteiger et al., 2020). Compared with SMILES strings, it is more natural to represent molecules as graphs and model them with Graph neural networks (GNNs). However, the training of GNNs requires a large amount of labeled molecule data and supervised-trained GNNs usually show limited generalization ability for newly synthesized molecules and new properties. In order to tackle these issues, self-supervised representation pre-training techniques are explored (Rong et al., 2020; Li et al., 2021; Stärk et al., 2022) in molecular property prediction.

**Self-supervised Learning of Graphs**   Based on how self-supervised tasks are constructed, previous works can be classified into two categories, contrastive models and predictive models. Contrastive models (Hu et al., 2019; Zhang et al., 2020; Sun et al., 2020; You et al., 2021; Sun et al., 2021; Subramonian, 2021; Xia et al., 2022; Li et al., 2022b; Suresh et al., 2021) generate different views for each graph via data augmentation and learn representations by contrasting the similarities between views of the same graph and different graphs. Predictive models (Hu et al., 2020; Rong et al., 2020; Hou et al., 2022) generally mask a part of the graph and predict the masked parts. Most existing methods focus on learning node-level or graph-level representations, with some work involving subgraph-level feature that utilizes the rich semantic information contained in the subgraphs or motifs. For instance, in (Zhang et al., 2021), the topology information of motifs is considered. In (Wu et al., 2023), a Transformer architecture is proposed to incorporate motifs and construct 3D heterogeneous molecular graphs for representation learning. Different from these works, we propose a bilateral fusion model with a novel subgraph-aware GNN branch and propose a joint node-wise and graph-wise self-supervised training strategy so that the learned representation can capture both atom-wise and subgraph-wise information.

## 3 METHODOLOGY

### 3.1 PROBLEM FORMULATION

We represent a molecule as a graph $G = <V, E>$ with node attribute vectors $\boldsymbol{x}_v$ for $v \in V$ and edge attribute vectors $\boldsymbol{e}_{uv}$ for $(u, v) \in E$, where $V$ and $E$ are the sets of atoms and bonds, respectively. We consider a multilabel classification problem with instance $G$ and label $\boldsymbol{y} = \{y_l\}_{l=1}^L$, where $y_l \in \{0, 1\}$ denotes whether this property is present in $G$. Given a set of training samples $\mathcal{D}_{train} = \{(G_i, \boldsymbol{y}_i)\}_{i=1}^{N_1}$, our target is to learn a mapping $f : G \to [0, 1]^L$ that can well generalize to the test set. We also have a set of unlabelled support set $\mathcal{D}_{support} = \{(G_i)\}_{i=1}^{N_2}$, where $N_2 >> N_1$, and apply our self-supervised learning method to get better initial representation.

### 3.2 ATOM-WISE BRANCH

Previous works extract the representation of a molecule by aggregating the embeddings of all atoms with GNNs. Similarly, our atom-wise branch applies a single GNN model with $K$ layers to map each molecule graph into an embedding. Specifically, for $G = <V, E>$, the input embedding $\boldsymbol{h}_v^0$ of the node $v \in V$ is initialized by $\boldsymbol{x}_v$, the input embedding at the $k$-th layer $\boldsymbol{e}_{uv}^k$ of the edge

$(u, v) \in E$ is initialized by $\boldsymbol{e}_{uv}$, and the $K$ GNN layers iteratively update $\boldsymbol{h}_v$ by polymerizing the embeddings of neighboring nodes and edges of $\hat{v}$. In the $k$-th layer, $\boldsymbol{h}_v^{(k)}$ is updated as follows:

$$\boldsymbol{h}_v^{(k)} = \text{COMBINE}^{(k)}(\boldsymbol{h}_v^{(k-1)}, \text{AGGREGATE}^{(k)}(\{(\boldsymbol{h}_v^{(k-1)}, \boldsymbol{h}_u^{(k-1)}, \boldsymbol{e}_{uv}^k) : u \in \mathcal{N}(v)\})) \quad (1)$$

where $\boldsymbol{h}_v^{(k)}$ denotes the embedding of node $v$ at the $k$-th layer, and $\mathcal{N}(v)$ represents the neighborhood set of node $v$. After $K$ iterations of aggregation, $\boldsymbol{h}_v^{(K)}$ captures the structural information within its $K$-hop network neighborhoods. The embedding $\boldsymbol{z}_A$ of the graph $G$ is the average of each node.

$$\boldsymbol{z}_A = \text{MEAN}(\{\boldsymbol{h}_v^{(K)} | v \in V\}) \quad (2)$$

Then we add a linear classifier $\boldsymbol{z}_A \to \mathbb{R}^L$. Formally, atom-wise architecture can be described as learning a mapping $f : G \to \mathbb{R}^L$. The loss function of the atom-wise architecture is:

$$\mathcal{L}_{atom} = \frac{1}{N_1} \sum_{(G,\boldsymbol{y}) \in \mathcal{D}_{train}} \ell\Big(f(G), \boldsymbol{y}\Big) \quad (3)$$

### 3.3 SUBGRAPH-WISE BRANCH

Atoms are influenced by their surrounding contexts and the semantics of a single atom can change significantly in different environments. Functional groups, which are connected subgraphs composed of coordinated atoms, determine many molecular properties. Our proposed hierarchical Decomposition-Polymerization architecture decouples the representation learning into the subgraph embedding phase, where each molecule is decomposed into subgraphs and an embedding vector is extracted from each subgraph, and the subgraph polymerization phase, where subgraphs are modeled as nodes and their embeddings are updated by polymerizing information from neighboring subgraphs. Finally, the final representation is obtained by combining all subgraph-wise embeddings.

**Subgraph vocabulary construction** Functional groups correspond to special subgraphs, however, pre-defined subgraph vocabularies of hand-crafted functional groups may be incomplete, i.e., not all molecules can be decomposed into disjoint subgraphs in the vocabulary. There exist many decomposition algorithms such as the principle subgraph extraction strategy (Kong et al., 2022) and breaking retrosynthetically interesting chemical substructures (BRICS) (Degen et al., 2008). Generally, we denote a subgraph of the molecule $G$ by $S = <\hat{V}, \hat{E}> \in G$, where $\hat{V}$ is a subset of $V$ and $\hat{E}$ is the subset of $E$ corresponding to $\hat{V}$. The target of principle subgraph extraction is to constitute a vocabulary of subgraphs $\mathbb{V} = \{S_{(1)}, S_{(2)}, \cdots, S_{(M)}\}$ that represents the meaningful patterns within molecules, where each unique pattern is associated with an index.

**Subgraph embedding** In this phase, we only focus on learning the embedding of each subgraph by modeling the intra-subgraph interactions. For a molecule $G = <V, E>$, we decompose it into a set of non-overlapped subgraphs $\{S_{\pi 1}, S_{\pi 2}, \cdots, S_{\pi T}\}$, where $T$ is the number of decomposed subgraphs and $\pi t$ is the corresponding index of the $t^{th}$ decomposed subgraph in the constructed vocabulary $\mathbb{V}$. For each subgraph $S_{\pi t} = <\hat{V}_{\pi t}, \hat{E}_{\pi t}>$, we have $\hat{V}_{\pi t} \subset V$ and $\hat{E}_{\pi t} \subset E$. For each edge $(u, v)$ in $E$, we add it into the inter-subgraph edge set $\mathcal{E}$ if it satisfies that nodes $u$ and $v$ are in different subgraphs. Therefore, we have $V = \cup \hat{V}_{\pi t}$ and $E = \cup \hat{E}_{\pi t} \cup \mathcal{E}$.

We apply a single GNN model with $K_1$ layers to map each decomposed subgraph into an embedding. GNN depends on the graph connectivity as well as node and edge features to learn an embedding for each node $v$. We discard the inter-subgraph edge set $\mathcal{E}$, any two subgraphs are disconnected and the information will be detached among subgraphs. This is equivalent to feeding each subgraph $S_{\pi t}$ into the GNN model individually.

By feeding the molecular graph after discarding all inter-subgraph edges into the GNN model, the embeddings of all atoms in the $T$ decomposed subgraphs are updated in parallel and the embeddings of all subgraphs can be obtained by adaptive pooling. Compared with previous strategies (Hu et al., 2019; Zhang et al., 2021) that directly obtain molecular representations from the context-dependent atom-wise embeddings with all edges, our strategy first extracts subgraph-level embeddings. When a subgraph appears in different molecules, both its atom-wise embeddings and the subgraph embedding remain the same.

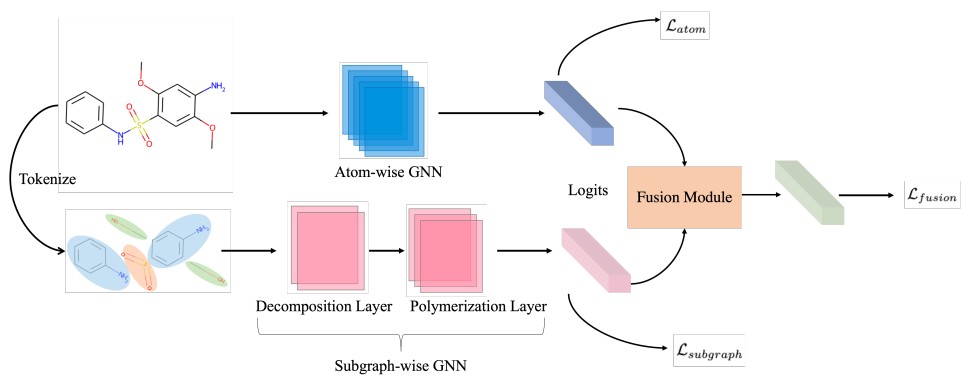

Figure 2: Overall architecture of our GSBF

**Subgraph-wise polymerization**  In the previous subgraph embedding phase, we view each atom in the subgraph as a node and extract the embedding of each subgraph. In the subgraph-wise polymerization phase, we polymerize the embeddings of neighboring subgraphs for acquiring representations of subgraphs and the final representation of the molecule $G$. Differently, we view each subgraph as a node and connect them by the set of inter-subgraph edges $\mathcal{E}$. Two subgraphs $S_{\pi t}$ and $S_{\pi l}$ are connected if there exists at least one edge $(\hat{u}, \hat{v}) \in \mathcal{E}$ where $\hat{u} \in \hat{V}_{\pi t}$ and $\hat{v} \in \hat{V}_{\pi l}$. In this way, we construct another graph whose nodes are subgraphs and employ another GNN model with $K_2$ layers to update the representation of each subgraph and extract the final representation $z_S$. At the $k'$-th layer, the embedding $h_{\pi t}$ for the $t$-th subgraph is updated as follows:

$$h_{\pi t}^{(k')} = \text{COMBINE}^{(k')}(h_{\pi t}^{(k'-1)}, \text{AGGREGATE}^{(k')}(\{h_{\pi t}^{(k'-1)}, h_{\pi l}^{(k'-1)}, e_{\hat{u}\hat{v}}^{k'}) : (\hat{u}, \hat{v}) \in \mathcal{E} \\ \text{AND} \quad \hat{u} \in \hat{V}_{\pi t} \quad \text{AND} \quad \hat{v} \in \hat{V}_{\pi l}\}) \tag{4}$$

As shown in Eq.5, representation $z_S$ has aggregated all information from different subgraphs, where $h_{\pi t}^{(K_2)}$ denotes the subgraph feature which is fed forward after $K_2$ iterations.

$$z_S = \text{MEAN}(\{h_{\pi t}^{(K_2)} | t \in \{1, 2, \cdots, k\}\}) \tag{5}$$

The semantics of subgraphs corresponding to functional groups are relatively more stable in different molecular structures. Our polymerization takes such subgraphs as basic units to model the structural interactions and geometric relationships between them. Similarly, we add a linear classifier $z_S \to \mathbb{R}^L$. Formally, subgraph-wise architecture can be described as learning a mapping $g : G \to \mathbb{R}^L$. The final loss function is:

$$\mathcal{L}_{subgraph} = \frac{1}{N_1} \sum_{(G, \boldsymbol{y}) \in \mathcal{D}_{train}} \ell\Big(g(G), \boldsymbol{y}\Big) \tag{6}$$

### 3.4 Gate-guided and Subgraph-aware Bilateral Fusion (GSBF)

The properties of some molecules are determined by their constituent atoms, while others are influenced by functional groups, and in most cases, it's a combination of both factors. When the property is more closely related to the atom, we should utilize more information from the atom-wise branch, and vice versa. The current fusion methods often directly aggregate information, which may not be sufficiently accurate (Wang et al., 2022; Fey et al., 2020). Therefore, we introduce our Gate-guided and Subgraph-aware Bilateral Fusion (GSBF) to automatically control the aggregation rate. Formally, we define the fusion feature as shown in Eq. 7, where *score* represents the aggregation rate and is defined in Eq. 8. Here, we denote $S(\cdot)$ as a sigmoid function, and $W_1$ and $W_2$ are two learnable parameters.

$$\tilde{z} = score_A \cdot z_A + score_S \cdot z_S + \frac{z_A + z_S}{2} \tag{7}$$

$$score_A = S(W_1^T X_1) \qquad score_S = S(W_2^T X_2) \tag{8}$$

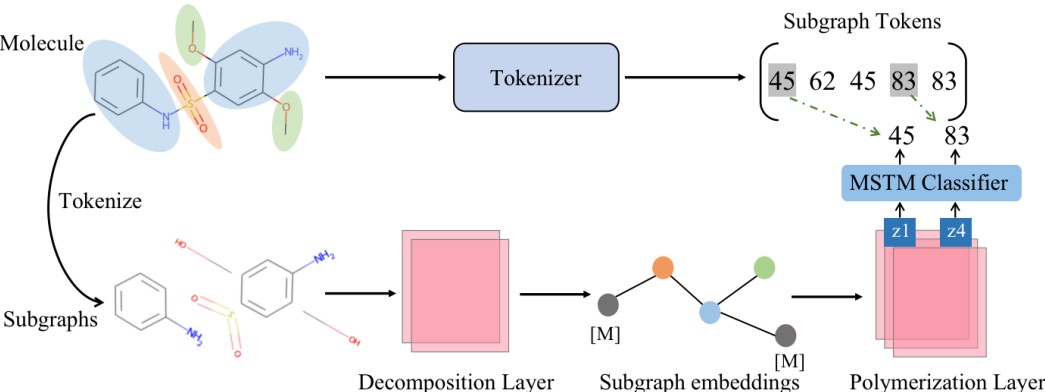

Figure 3: Overview of the MSTM. We first construct a subgraph vocabulary by automatically mining principle subgraphs from data and each subgraph in the vocabulary is tokenized. During pre-training, each molecule is decomposed into subgraphs and then fed into the decomposition module, which maps each subgraph into an embedding. Some subgraphs are randomly masked and replaced with a special embedding. These subgraph embeddings are modeled as nodes and further fed into the polymerization module to extract subgraph-level representations. The pre-training task is to predict the masked tokens from the corrupted subgraph-level representations with a linear predictor.

In our approach, the ultimate output feature, denoted as $\tilde{z}$, combines the invariant features from both branches, incorporating the weighted contributions from both the atom-wise and subgraph-wise branches. Typically, when a property exhibits a stronger association with subgraphs, the value of $score_S$ surpasses that of $score_A$, and conversely for properties leaning towards atoms. Our gate mechanism adeptly automates the feature selection process, effectively enhancing overall performance. Finally, we add a linear classifier $h : \tilde{z} \rightarrow \mathbb{R}^L$ to realize the classification task.

$$\mathcal{L}_{fusion} = \frac{1}{N_1} \sum_{(G, \boldsymbol{y}) \in \mathcal{D}_{train}} \ell\Big(h(\tilde{z}), \boldsymbol{y}\Big) \qquad (9)$$

### 3.5 SELF-SUPERVISED LEARNING

**Node-level self-supervised learning**  Many recent works show that self-supervised learning can learn generalizable representations from a large number of unlabelled molecules. Since the atom-wise branch and subgraph-wise branch are decoupled, we can apply the existing atom-wise self-supervised learning method to the atom-wise branch of GSBF such as attrMasking (Hu et al., 2019).

For the subgraph-wise branch, we propose the Masked Subgraph-Token Modeling (MSTM) strategy, which randomly masks some percentage of subgraphs and then predicts the corresponding subgraph tokens. As shown in Fig.3, a training molecule $G$ is decomposed into $T$ subgraphs $\{S_{\pi 1}, S_{\pi 2}, \cdots, S_{\pi T}\}$. The subgraphs are tokenized to tokens $\{\pi 1, \pi 2, \cdots \pi T\}$, respectively. Similar to BEiT (Bao et al., 2021), we randomly mask a number of $M$ subgraphs and replace them with a learnable embedding. Therefore, we construct a corrupted graph $\tilde{G}$ and feed it into our hierarchical decomposition-polymerization GNN architecture to acquire polymerized representations of all subgraphs. For each masked subgraph $\tilde{S}_{\pi t}$, we bring an MSTM classifier $p(\cdot | h_{\pi t})$ with weight $\boldsymbol{W}_p$ and bias $\boldsymbol{b}_p$ to predict the ground truth token $\pi t$. Formally, the pre-training objective of MSTM is to minimize the negative log-likelihood of the correct tokens given the corrupted graphs.

$$\mathcal{L}_{MSTM} = \frac{1}{N_2} \sum_{\tilde{G} \in \mathcal{D}_{support}} -\mathbb{E}\left[\sum_t \log p_{MSTM}(\pi t | g(\tilde{G}))\right] \qquad (10)$$

where $p_{MSTM}(\pi t | \tilde{G}) = Softmax(\boldsymbol{W}_p \tilde{\boldsymbol{h}}_{\pi t}^{(K_2)} + \boldsymbol{b}_p)$. Different from previous strategies, which randomly mask atoms or edges to predict the attributes, our method randomly masks some subgraphs and predicts their indices in the vocabulary $\mathbb{V}$ with the proposed decomposition-polymerization

Table 1: Test ROC-AUC performance of different methods on molecular property classification tasks. AVG represents the average results overall benchmarks. We highlight the best and second-best results with ∗ and ∗. We report the mean and standard results.

| Methods | BACE | BBBP | ClinTox | HIV | MUV | SIDER | Tox21 | ToxCast | Avg |
|---|---|---|---|---|---|---|---|---|---|
| Infomax | 75.9(1.6) | 68.8(0.8) | 69.9(3.0) | 76.0(0.7) | 75.3(2.5) | 58.4(0.8) | 75.3(0.5) | 62.7(0.4) | 70.3 |
| AttrMasking | 79.3(1.6) | 64.3(2.8) | 71.8(4.1) | 77.2(1.1) | 74.7(1.4) | 61.0(0.7) | 76.7(0.4) | 64.2(0.5) | 71.1 |
| GraphCL | 75.4(1.4) | 69.7(0.7) | 76.0(2.7) | 78.5(1.2) | 69.8(2.7) | 60.5(0.9) | 73.9(0.7) | 62.4(0.6) | 70.8 |
| AD-GCL | 78.5(0.8) | 70.0(1.1) | 79.8(3.5) | 78.3(1.0) | 72.3(1.6) | 63.3(0.8) | 76.5(0.8) | 63.1(0.7) | 72.7 |
| MGSSL | 79.1(0.9) | 69.7(0.9) | 80.7(2.1) | 78.8(1.2) | 78.7(1.5) | 61.8(0.8) | 76.5(0.3) | 64.1(0.7) | 73.7 |
| GraphLoG | 83.5(1.2) | 72.5(0.8) | 76.7(3.3) | 77.8(0.8) | 76.0(1.1) | 61.2(1.1) | 75.7(0.5) | 63.5(0.7) | 73.4 |
| GraphMVP | 81.2(0.9) | 72.4(1.6) | 77.5(4.2) | 77.0(1.2) | 75.0(1.0) | 63.9(1.2) | 74.4(0.2) | 63.1(0.4) | 73.1 |
| GraphMAE | 83.1(0.9) | 72.0(0.6) | 82.3(1.2) | 77.2(1.0) | 76.3(2.4) | 60.3(1.1) | 75.5(0.6) | 64.1(0.3) | 73.8 |
| Mole-Bert | 80.8(1.4) | 71.9(1.6) | 78.9(3.0) | 78.2(0.8) | 78.6(1.8) | 62.8(1.1) | 76.8(0.5) | 64.3(0.2) | 74.0 |
| GSBF | 80.7(1.6) | 74.3(0.4) | 80.9(2.1) | 77.0(0.8) | 77.1(1.1) | 62.9(0.8) | 76.0(0.2) | 65.5(0.3) | 74.3 |

Table 2: Test RMSE performance of different methods on the regression datasets.

| Methods | Regression dataset | | | | | |
|---|---|---|---|---|---|---|
| | fine-tuning | | | linear probing | | |
| | FreeSolv | ESOL | Lipo | FreeSolv | ESOL | Lipo |
| Infomax | 3.416(0.928) | 1.096(0.116) | 0.799(0.047) | 4.119(0.974) | 1.462(0.076) | 0.978(0.076) |
| EdgePred | 3.076(0.585) | 1.228(0.073) | 0.719(0.013) | 3.849(0.950) | 2.272(0.213) | 1.030(0.024) |
| Masking | 3.040(0.334) | 1.326(0.115) | 0.724(0.012) | 3.646(0.947) | 2.100(0.040) | 1.063(0.028) |
| ContextPred | 2.890(1.077) | 1.077(0.029) | 0.722(0.034) | 3.141(0.905) | 1.349(0.069) | 0.969(0.076) |
| GraphLog | 2.961(0.847) | 1.249(0.010) | 0.780(0.020) | 4.174(1.077) | 2.335(0.073) | 1.104(0.024) |
| GraphCL | 3.149(0.273) | 1.540(0.086) | 0.777(0.034) | 4.014(1.361) | 1.835(0.111) | 0.945(0.024) |
| GraphMVP | 2.874(0.756) | 1.355(0.038) | 0.712(0.025) | 2.532(0.247) | 1.937(0.147) | 0.990(0.024) |
| GSBF | 2.789(0.739) | 0.917(0.080) | 0.541(0.014) | 3.001(0.731) | 1.257(0.200) | 0.805(0.051) |

architecture. Actually, our prediction task is more difficult since it operates on subgraphs and the size of $\mathbb{V}$ is larger than the size of atom types. As a result, the learned substructure-aware representation captures high-level semantics of substructures and their interactions and can be better generalized to the combinations of known subgraphs under different scaffolds.

**Graph-level self-supervised learning**  Node-level pre-training alone is insufficient for obtaining features that can be generalized (Xia et al., 2023). Therefore, we propose graph-level self-supervised learning, as illustrated in Eq. 11, where $\mathcal{B}^-$ represents negative samples for the anchor sample $G_i$. These negative samples are comprised of the remaining samples within the same batch. We define $\boldsymbol{v}_i = (\boldsymbol{z}_{A_i} + \boldsymbol{z}_{S_i})/2$, and $(\boldsymbol{v}_i, \boldsymbol{v}_i')$ constitutes a pair of augmentation graphs derived from $G_i$. In the atom-wise branch, we randomly remove some atoms, and in the subgraph-wise branch, we randomly remove one subgraph to implement augmentation.

$$\mathcal{L}_{cl} = \frac{1}{N_2} \sum_{G_i \in \mathcal{D}_{support}} - \log \frac{\exp\left(\boldsymbol{v}_i \cdot \boldsymbol{v}_i'\right)}{\exp\left(\boldsymbol{v}_i \cdot \boldsymbol{v}_i'\right) + \sum_{G_j \in \mathcal{B}^-} \exp\left(\boldsymbol{v}_i \cdot \boldsymbol{v}_j\right)} \tag{11}$$

Our method is different from GraphCL (You et al., 2020) and Mole-Bert Xia et al. (2023), which apply graph-level augmentation on the atom-wise branch only. Our method maximizes the feature invariance along these two branches and improves the generalization for downstream tasks. In addition, graph-level self-supervised learning makes the two branches interact which can utilize the superiority of our bilateral architecture.

## 4 EXPERIMENTS

### 4.1 DATASETS AND EXPERIMENTAL SETUP

**Datasets and Dataset Splittings**  We use the ZINC250K dataset (Sterling & Irwin, 2015) for self-supervised pre-training, which is constituted of $250k$ molecules up to 38 atoms. As for downstream

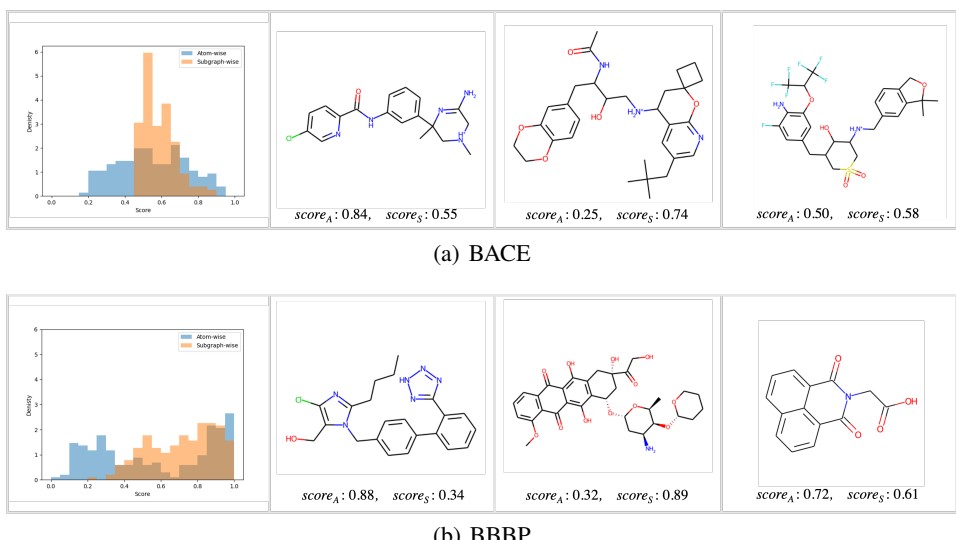

Figure 4: Visualization of our gate mechanism. (a) The probability density and some molecules on the BACE dataset. (b) The probability density and some molecules on the BBBP dataset.

molecular property prediction tasks, we test our method on 8 classification tasks and 3 regression tasks from MoleculeNet (Wu et al., 2018). For classification tasks, we follow the *scaffold-splitting* (Ramsundar et al., 2019), where molecules are split according to their scaffolds (molecular substructures). The proportion of the number of molecules in the training, validation, and test sets is $80\% : 10\% : 10\%$. Following (Li et al., 2022a), we apply random scaffold splitting to regression tasks, where the proportion of the number of molecules in the training, validation, and test sets is also $80\% : 10\% : 10\%$. Following (Zhang et al., 2021; Liu et al., 2021), we performed 10 replicates on each dataset to obtain the mean and standard deviation.

**Baselines** For classification tasks, we comprehensively evaluated our method against different self-supervised learning methods on molecular graphs, including Infomax (Veličković et al., 2018), AttrMasking (Hu et al., 2019), ContextPred (Hu et al., 2019), GraphCL (You et al., 2020), AD-GCL (Suresh et al., 2021), MGSSL (Zhang et al., 2021), GraphLog (Xu et al., 2021), Graph-MVP (Liu et al., 2021), GraphMAE (Hou et al., 2022), and Mole-Bert Xia et al. (2023). For regression tasks, we compare our method with Infomax (Veličković et al., 2018), EdgePred (Hamilton et al., 2017), AttrMasking (Hu et al., 2019), ContextPred (Hu et al., 2019), GraphLog (Xu et al., 2021), GraphCL (You et al., 2020),and GraphMVP (Liu et al., 2021). Among them, 3DInfomax exploits the three-dimensional structure information of molecules, while other methods also do not use knowledge or information other than molecular graphs.

## 4.2 RESULTS AND ANALYSIS

**Classification** Table 1 presents the results of fine-tuning compared with the baselines on classification tasks. "GSBF" denotes the results of our method after self-supervised per-training. From the results, we observe that the overall performance of our method is significantly better than all baseline methods on most datasets. Among them, AttrMasking and GraphMAE also use masking strategies that operate on atoms and bonds in molecular graphs. Compared with AttrMasking, our method achieves a significant performance improvement of 10.0%, 9.1%, and 2.4% on BBBP, ClinTox, and MUV datasets respectively, with an average improvement of 3.2% on all datasets. Compared with GraphMAE, our method also achieved a universal improvement. Compared with contrastive learning models, our method achieves a significant improvement with an average improvement of 4.0% compared with Infomax, 3.5% compared with GraphCL, 1.6% compared with AD-GCL, and 0.9% compared with GraphLoG. For GraphMVP which combines contrastive and generative methods, our method also has an average improvement of 1.2%.

Table 3: Ablation study on different components of our dual branch self-supervised learning method.

| Methods | BACE | BBBP | ClinTox | HIV | MUV | SIDER | Tox21 | ToxCast | Avg |
|---|---|---|---|---|---|---|---|---|---|
| Node-level | 78.8(1.9) | 72.8(1.4) | 74.2(2.8) | 77.1(0.6) | 73.7(1.5) | 61.3(0.9) | 75.9(0.5) | 65.6(0.3) | 72.4 |
| Graph-level | 73.9(1.4) | 69.5(2.3) | 61.8(3.2) | 75.6(1.0) | 73.1(1.9) | 59.0(0.8) | 74.6(0.3) | 63.1(0.5) | 68.8 |
| Node+Graph | 80.7(1.6) | 74.3(0.4) | 80.9(2.1) | 77.0(0.8) | 77.1(1.1) | 62.9(0.8) | 76.0(0.2) | 65.5(0.3) | 74.3 |

**Regression**    In Table 2, we report evaluation results in regression tasks under the fine-tuning and linear probing protocols for molecular property prediction. Other methods are pre-trained on the large-scale dataset ChEMBL29 (Gaulton et al., 2012) containing 2 million molecules, which is 10 times the size of the dataset for pre-training our method. The comparison results show that our method outperforms other methods and achieves the best performance in five out of six tasks, despite being pre-trained only on a small-scale dataset. This indicates that our method can better learn transferable information about atoms and subgraphs from fewer molecules with higher data-utilization efficiency.

**The effectiveness of our self-supervised learning in the pre-training stage**    From Table 1 and Table 2, it is evident that our self-supervised learning during the pre-training stage yields superior results in most tasks. Our self-supervised learning approach comprises node-level and graph-level learning components, and we conduct an independent analysis of their effectiveness. The experimental results presented in Table 3 indicate that joint pre-training of the two branches leveraging two self-supervised learning methods is more effective than pre-training separately (i.e., solely applying node-level or graph-level self-supervised components). To elaborate, combining both self-supervised learning components results in a 1.9% improvement compared to using node-level mask reconstruction alone and a 5.5% improvement compared to using graph-level contrastive learning alone. These findings underscore the significance of combining these two self-supervised learning components and facilitating interaction between the two branches.

**Visualization of our Gate-guided Module**    In Fig. 4, we illustrate the distribution of Atom-wise ($score_A$) and Subgraph-wise ($score_S$) on the BACE and BBBP datasets, respectively. In the case of the BACE dataset, the values of $score_A$ exhibit a uniform distribution, whereas $score_S$ is predominantly clustered around 0.5. Conversely, for the BBBP dataset, the distribution of $score_A$ follows a similar binomial pattern, which complements the distribution of $score_S$. This observation highlights that our gate mechanism is capable of discerning the significance of subgraphs and atoms in an adaptative manner. Additionally, we provide visual representations of selected instances along with their corresponding score values. Typically, molecules with a greater number of atoms tend to allocate more attention to the subgraph-wise branch, while those with fewer atoms prioritize the atom-wise branch. This preference arises due to the increased complexity of message passing as the number of atoms grows. Within the subgraph-wise branch, we reestablish connections between different subgraphs and enhance interactions among them. Consequently, our subgraph-wise branch tends to provide more benefits to molecules with a higher number of atoms.

## 5    CONCLUSION

In this paper, we consider the fact that molecular properties are not solely determined by either atoms or subgraphs and introduce a novel approach called Gate-guided and Subgraph-aware Bilateral Fusion (GSBF). The GSBF model comprises two branches: one for modeling atom-wise information and the other for subgraph-wise information. Furthermore, recognizing that properties are determined differently by atoms or subgraphs, we propose a gate-guided mechanism to automatically fuse the features of these two branches. To enhance generalization, we introduce a node-level self-supervised learning method called MSTM for the less-explored subgraph-wise branch. Additionally, we introduce a graph-level self-supervised learning method to maximize the average invariance of the two branches. Experimental results verify the effectiveness of our approach.

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

Table 4: Test ROC-AUC performance of different methods on molecular property classification tasks with different tokenization algorithms and model configurations.

| Methods | BACE | BBBP | ClinTox | HIV | MUV | SIDER | Tox21 | ToxCast | Avg |
|---|---|---|---|---|---|---|---|---|---|
| Atom-wise | 71.6(4.5) | 68.7(2.5) | 57.5(3.8) | 75.6(1.4) | 73.2(2.5) | 57.4(1.1) | 74.1(1.4) | 62.4(1.0) | 67.6 |
| The principle subgraph, $|\mathbb{V}| = 100, K_1 = 2, K_2 = 3$ | | | | | | | | | |
| Subgraph-wise | 64.4(5.2) | 69.4(3.0) | 59.2(5.2) | 71.7(1.5) | 68.3(1.6) | 59.1(0.9) | 72.6(0.8) | 61.3(0.7) | 65.8 |
| GSBF | 72.1(2.8) | 72.4(2.3) | 57.0(4.8) | 74.7(1.9) | 71.4(1.8) | 59.6(1.6) | 76.1(0.8) | 63.8(0.6) | 68.4 |
| The principle subgraph, $|\mathbb{V}| = 100, K_1 = 3, K_2 = 2$ | | | | | | | | | |
| Subgraph-wise | 63.1(7.1) | 68.7(2.7) | 55.8(6.3) | 71.7(1.7) | 68.3(4.9) | 58.7(1.5) | 72.9(0.9) | 61.5(0.9) | 65.1 |
| GSBF | 73.4(3.3) | 70.7(2.9) | 59.5(4.2) | 74.8(1.1) | 71.8(2.8) | 59.2(1.7) | 75.8(0.7) | 64.4(0.6) | 68.7 |
| The principle subgraph, $|\mathbb{V}| = 300, K_1 = 2, K_2 = 3$ | | | | | | | | | |
| Subgraph-wise | 66.2(4.5) | 63.7(3.2) | 59.0(8.5) | 74.2(1.6) | 68.9(1.9) | 61.6(1.8) | 73.3(0.9) | 60.5(0.5) | 65.9 |
| GSBF | 69.3(4.5) | 67.3(4.1) | 62.7(5.2) | 76.2(2.1) | 72.2(2.9) | 60.1(2.1) | 76.0(0.8) | 63.7(0.7) | 68.4 |
| The principle subgraph, $|\mathbb{V}| = 300, K_1 = 3, K_2 = 2$ | | | | | | | | | |
| Subgraph-wise | 67.3(2.0) | 66.5(3.3) | 54.7(5.7) | 73.8(2.1) | 69.7(2.5) | 60.8(2.5) | 73.7(0.7) | 60.8(0.7) | 65.9 |
| GSBF | 74.8(2.8) | 69.4(2.9) | 57.0(3.9) | 77.1(0.9) | 72.5(2.2) | 60.5(1.6) | 75.9(0.7) | 63.8(0.7) | 68.9 |
| BRICS, $K_1 = 2, K_2 = 3$ | | | | | | | | | |
| Subgraph-wise | 71.4(3.9) | 66.1(3.5) | 51.8(3.7) | 75.2(1.8) | 70.0(2.0) | 56.0(1.5) | 74.0(0.8) | 64.2(1.1) | 66.1 |
| GSBF | 72.4(4.3) | 69.6(1.8) | 60.8(6.7) | 75.9(1.5) | 73.3(2.4) | 58.1(1.3) | 75.8(0.6) | 65.5(0.7) | 68.9 |
| BRICS, $K_1 = 3, K_2 = 2$ | | | | | | | | | |
| Subgraph-wise | 73.6(3.7) | 67.0(1.9) | 53.0(5.3) | 74.0(1.7) | 70.5(1.9) | 55.8(1.7) | 74.4(1.0) | 65.0(0.4) | 66.7 |
| GSBF | 72.4(4.6) | 69.9(1.8) | 53.5(11.5) | 77.0(1.0) | 73.9(1.9) | 55.7(1.5) | 75.7(0.8) | 64.9(1.1) | 67.9 |

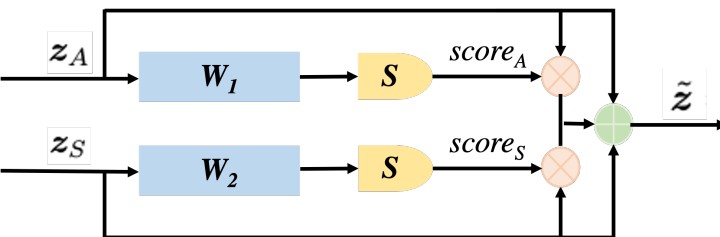

Figure 5: The internal structure of our Fusion Module.

# A APPENDIX

## A.1 ARCHITECTURE OF THE FUSION MODULE

Fig. 5 shows the internal structure of our Fusion Module. The output feature of the atom-wise branch ($z_A$) and the output feature of the subgraph-wise branch ($z_S$) are fed into a learnable module to get the score, which is normalized by sigmoid function. For the final output feature, we fuse the re-weight features and original output features.

## A.2 GSBF CAN ACHIEVE BETTER GENERALIZATION

In Table 4, we compare the performance of our atom-level, subgraph-level, and integrated GSBF model on different classification tasks. From the experimental results, it can be seen that the atom-level branch performs better than the subgraph-level branch on some datasets, such as BACE, Tox-Cast, and Tox21, while the subgraph-level branch outperforms on others, such as SIDER and BBBP. This is because the influencing factors of different classification tasks are different, some focus on functional groups, while some focus on the interactions between atoms and chemical bonds. However, no matter how the parameters of the model change, our GSBF always achieves better results than the two separate branches on all datasets since it adaptively integrates the strengths of both. These results demonstrate that our GSBF has better generalization ability.

A.3 TOKENIZATION ALGORITHM

We do experiments with different tokenization algorithms and we roughly introduce these methods in this section.

**Introduction of BRICS algorithm**    The BRICS algorithm is one of the molecular fragmentation methods, which is an improved and optimized algorithm based on the RECAP algorithm. The BRICS algorithm introduces a better set of fragmentation rules and a set of recombinant motifs rules to form a fragmentation space. Specifically, the BRICS algorithm obtains active building blocks by segmenting active molecules. It is known that some common chemical reactions form bonds, so when segmenting molecules, BRICS segments these bonds. The algorithm splits into 16 pre-defined bonds. These 16 pre-defined bonds ensure that the split fragments are suitable for combination and applicable to combinatorial chemistry, and these 16 pre-defined bonds are given in the form of fragment structures. When segmenting molecules, all breakable bonds are cut off at the same time to avoid redundant fragments, and if the fragments after cleavage only contain small functional groups (such as hydrogen, methyl, ethyl, propyl, and butyl), the fragments will not be cleaved again to avoid generating useless small fragments. In the process of splitting, the algorithm preserves the cyclic structure. After each bond is broken, two breakpoints are formed. RECAP algorithm directly annotates 'isotope labels' at the divided breakpoints, that is, the ids of breakable bonds, but BRICS divides firstly annotates "isotope labels" at the breakpoints, and then replaces these isotope labels with link atoms. For the RECAP algorithm, the ids of breakable bonds corresponding to the isotope labels annotated at the two breakpoints are the same, but for the BRICS algorithm, they are different, which also proves that the BRICS algorithm takes the chemical environment and surrounding substructures of each broken bond into account, and the partition effect is better. Finally, the molecules decomposed by the BRICS algorithm are a list composed of a one-step partitioned non-redundant fragment string.

**The principle subgraph extraction**    Principal Subgraph Extraction is a method of molecular fragmentation (Kong et al., 2022). It is mainly divided into three steps: initialization, merging, and updating. Before starting, the number of subgraphs to be divided $N$ is preset. First, the subgraphs are initialized as individual unique atoms, then two adjacent fragments are merged each time to form a new set of subgraphs, and the subgraph with the highest frequency in this step is selected as the newly divided subgraph. The merging and selection steps are repeated until the number of subgraphs divided reaches the preset number $N$. It should be noted that the definition of adjacent fragments here is fragments containing at least one first-order adjacent node of the node.

A.4 MODEL CONFIGURATION AND IMPLEMENTED DETAILS

**Supervised learning setting**    Our method involves two branches and our final loss function is shown as follows.

$$\mathcal{L}_{SL} = \mathcal{L}_{atom} + \mathcal{L}_{subgraph} + \mathcal{L}_{fusion} \tag{12}$$

The reason we retain $\mathcal{L}_{atom}$ and $\mathcal{L}_{subgraph}$ is that we would like to preserve the discriminative ability for the features of atom-wise and subgraph-wise independently while keeping the fusion feature discriminative at the same time. In the inference phase, we only use the fusion branch to get the prediction score.

**Self-supervised learning setting**    Our method involves node-level self-supervised learning and graph-level self-supervised learning. The final loss function is as follows.

$$\mathcal{L}_{SSL} = \mathcal{L}_{MSTM} + \mathcal{L}_{AttrMasking} + \mu \mathcal{L}_{cl} \tag{13}$$

We denote $\mathcal{L}_{AttrMasking}$ as the Attribute Masking method for the atom-wise branch. We set $\mu = 0.1$ in our experiments since we give more importance to mask reconstruction to learn the representation of two branches and contrastive learning aims to interact the two branches.

**Implemented details**    To verify the effectiveness of our GSBF, we do experiments with different molecular fragmentation methods, such as BRICS (Degen et al., 2008) and the principle subgraph (Kong et al., 2022). We also do experiments with different hyper-parameters $K_1$ and $K_2$. For self-supervised learning, we employ experiments using the principle subgraph with vocabulary

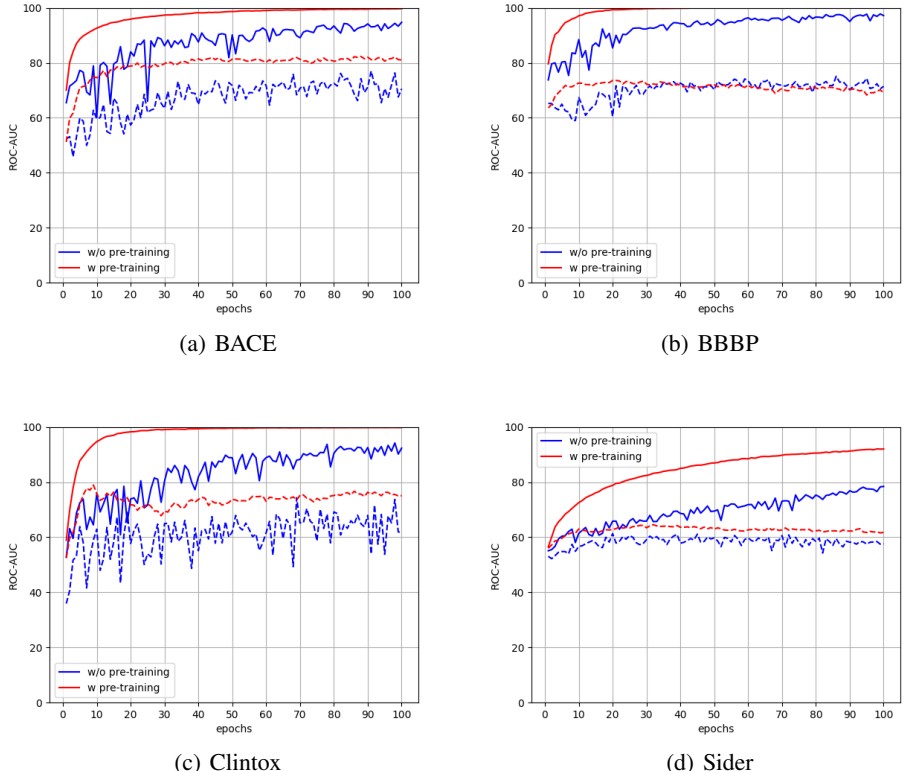

Figure 6: Training and testing curves. The solid lines denote training curves and the dashed lines denote testing curves. Our method shows better convergence and generalization.

size $|\mathbb{V}| = 100$. We perform the subgraph embedding module and the subgraph-wise polymerization module with a $K_1 = 2$ layer GIN (Leskovec & Jegelka, 2019) and a $K_2 = 3$ layer GIN, respectively. For downstream classification tasks and regression tasks, we mainly follow previous works (Hu et al., 2019) and (Li et al., 2022a), respectively. For the pre-training phase, we follow the work (Zhang et al., 2021) to use the Adam optimizer (Kingma & Ba, 2014) with a learning rate of $1 \times 10^{-3}$, and batch size is set to 32.

## A.5 VISUALIZATION CURVES

We also visualize the training and testing curves of our method. As shown in Fig. 6, our method achieves faster convergence compared with the model without pre-training. Moreover, our method achieves better generalization and the performances on test sets are consistently better than those of the model without pre-training throughout the whole training process.

## A.6 ERROR BARS

We also visualize the error bars in Fig. 7. For small datasets like Bace and Clintox, there exists heavy uncertainty. It is shown that the uncertainty of the subgraph-wise branch is only smaller than the atom-wise branch but achieves lower performance. Our method fuses the atom-wise and subgraph-wise information and can explicitly reduce the uncertainty and achieve better performance at the same time.

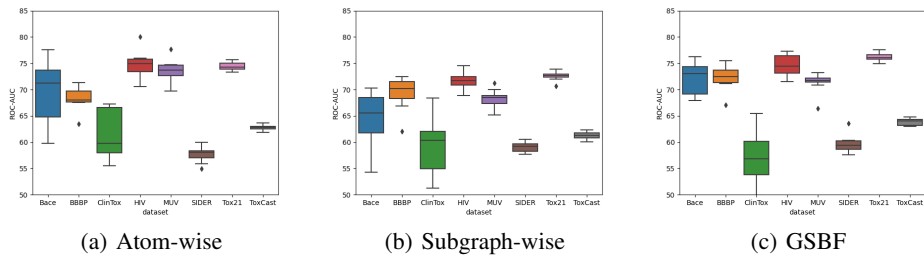

(a) Atom-wise       (b) Subgraph-wise       (c) GSBF

Figure 7: Error bars for (a) the atom-wise branch, (b) the subgraph-wise branch, and (c) GSBF.

Table 5: Ablation study on the gate mechanism.

| Methods | BACE | BBBP | ClinTox | HIV | MUV | SIDER | Tox21 | ToxCast | Avg |
|---------|------|------|---------|-----|-----|-------|-------|---------|-----|
| GSBF(w/o gate) | 77.1(3.2) | 73.8(0.8) | 80.4(3.0) | 77.2(0.8) | 77.6(0.5) | 62.8(0.6) | 75.9(0.4) | 65.2(0.3) | 73.8 |
| GSBF | 80.7(1.6) | 74.3(0.4) | 80.9(2.1) | 77.0(0.8) | 77.1(1.1) | 62.9(0.8) | 76.0(0.2) | 65.5(0.3) | 74.3 |

## A.7 EFFECTIVENESS OF GATE MECHANISM

There exists many fusion mechanisms such as fusing the features from atom-wise branch and subgraph-wise branch directly. To verify the effectiveness of our method, we do additional experiments, and the results are shown in Table 5. We denote GSBF(w/o gate) as fusing the features of two branches directly. The results verify the effectiveness of our gate mechanism.

