# OpenReview forum: "Gate-guided and subgraph-aware Bilateral Fusion for Molecular Property Prediction"
_ICLR.cc/2024/Conference — ICLR 2024 Conference Withdrawn Submission_

### Official Review · Reviewer_7exA · 2023-10-17

**Soundness:** 2 fair
**Presentation:** 3 good
**Contribution:** 2 fair
**Rating:** 5
**Confidence:** 4

**Summary:**

The authors propose to combine both subgraph-wise and atom-wise information with a two-branch model and a gate mechanism for molecular representation learning. In the subgraph-wise branch, the molecular graph is decomposed into subgraph nodes, while in the atom-wise branch, each node is an atom. Furthermore, the authors propose to pretrain the neural network with mask-predict paradigm on both two branches, as well as contrastive learning between global representations obtained by the two branches. Experiments on molecular property prediction demonstrate the proposed method can effectively capture bilateral information.

**Strengths:**

1. The proposed method utilizes data more efficiently as it still achieves overall better performance on the regression tasks with a pretraining dataset 10 times smaller than the ones used by baselines.
2. The gate mechanism can adaptively decide the preferred branch by learning from the data. The authors also analyzed the pattern of the gate mechanism to provide some interpretability.

**Weaknesses:**

1. For the pretraining part, both of the mask-predict paradigm and the contrastive learning have been well explored by the community as mentioned in the paper, which may limit the technical novelty of the proposed method.
2. Ablations on two-branch mask-predict pretraining (i.e. subgraph-only or atom-only pretrain) as well as the performance of the from-scratch baseline are not provided, which are necessary for identifying the importance of pretraining on both branches and the absolute gains of pretraining.

**Questions:**

1. In figure 1 (b), the curve shows BBBP is more correlated to the subgraph branch. However, in figure 1(c), it looks like on Tox21 the atom-wise branch and the subgraph-branch have similar performance, which can not support the claim that some properties are more correlated to atom-wise information.
2. What is $X_1$ and $X_2$ in equation (8)?
3. It is better to mention in the main text that some ablations are in the appendix (e.g. the gate mechanism) because they are important for identifying the importance of different modules.

---

### Official Review · Reviewer_ni8k · 2023-10-23

**Soundness:** 2 fair
**Presentation:** 1 poor
**Contribution:** 2 fair
**Rating:** 3
**Confidence:** 5

**Summary:**

The paper introduces the Gate-guided and Subgraph-aware Bilateral Fusion (GSBF) model. Unlike traditional methods that focus solely on atom-centric properties, GSBF seamlessly integrates both atom-wise and subgraph-wise information. It employs a unique gate-guided system to ensure better feature extraction. Additionally, the paper proposes a Masked Subgraph-Token Modeling strategy for self-supervised learning.

**Strengths:**

1. The research topic is important.
2. Bilateral Fusion of atom-wise and subgraph-wise information is interesting.

**Weaknesses:**

1. My first concern is about the paper's literature review, as it appears incomplete. The concept of merging atom-wise and subgraph-wise data isn't novel in our field. Works such as [1] has previously employed this combination for predicting molecular properties. [2] employs information at the atom-level, motif-level, and graph-level to develop a hierarchical model tailored for self-supervised learning. It's imperative for the authors to acknowledge these references and elucidate the primary distinctions between their approach and the methods proposed in these literatures.
2. The motivation of the paper is atom-wise and subgraph-wise models cannot accurately predict molecular properties independently. And the main contribution of the paper is proposing a model GSBF to address this dilemma. However, before demonstrating the effectiveness of GSBF, the authors spend almost all experimental sections on self-supervised learning setup. I assume that it is because authors also introduce a MSTM strategy for self-supervised learning. However, given that the method section's main content revolves around GSBF, I would strongly advise the authors to initially highlight the effectiveness of GSBF prior to delving into MSTM.
3. Given that GSBF serves as the foundational model for MSTM, and considering that other baseline methods like Informax and AttrMasking are built upon GIN, it would be highly beneficial for the authors to include a version of GSBF without pre-training as a comparison baseline.

[1]. Yu, Zhaoning, and Hongyang Gao. "Molecular representation learning via heterogeneous motif graph neural networks." International Conference on Machine Learning. PMLR, 2022.

[2].  Zang, Xuan, Xianbing Zhao, and Buzhou Tang. "Hierarchical molecular graph self-supervised learning for property prediction." Communications Chemistry 6.1 (2023): 34.

**Questions:**

Please refer to Weaknesses section.

---

### Official Review · Reviewer_P8ct · 2023-10-31

**Soundness:** 2 fair
**Presentation:** 3 good
**Contribution:** 1 poor
**Rating:** 1
**Confidence:** 5

**Summary:**

The paper studies 2D molecular graph representation learning. The overall learning framework contains two branches; a node-wise learning branch and a subgraph-wise branch. Finally, two branches are fused together with a learnable gate guiding the importance of each branch. Particularly, subgraphs are indexed from a constructed vocabulary, and they are then treated as super-nodes of a graph as input to GNNs.

**Strengths:**

Two branches are designed for molecular graph learning, and self-supervised learning fashion is employed for joint learning of two branches. Performance is good on several tasks.

**Weaknesses:**

1. The authors omit an important work [1] for molecular representation learning, where two branches are also used; one is atom-wise learning and the other is motif-wise learning. Motifs are subgraphs also indexed from a vocabulary. Given the high technical similarity with this existing work [1], the novelty of the paper is very limited. Also, the authors don't even mention this work, which is surprising given its high relevance.

2. Designing two branches is straightforward. In each branch, there are no shining ideas or points, which again weakens the technical contribution of this paper.

3. Self-supervised fashion for different views of a molecular graph is not new, and the authors should clearly state what's new in this paper.

4. The performance on 8 classification tasks is generally weak, with the best results on 2 tasks. Considering the efficiency issues caused by two branches and the fusion (no clear discussion regarding efficiency in the paper though), I would question how useful the method could be in practice.

Overall, the paper is far below the bar of ICLR.

[1]. Yu, Zhaoning, Molecular Representation Learning via Heterogeneous Motif Graph Neural Networks, ICML2022

**Questions:**

See weakness. I suggest the authors revise the paper significantly to consider existing works and clearly state the differences and novelty of this paper.

---

### Official Review · Reviewer_KxV7 · 2023-11-01

**Soundness:** 3 good
**Presentation:** 2 fair
**Contribution:** 2 fair
**Rating:** 3
**Confidence:** 5

**Summary:**

In the presented paper, the authors introduce a novel Graph Neural Network (GNN) approach named Gate-guided and Subgraph-aware Bilateral Fusion (GSBF) for predicting molecular properties. The proposed method takes into consideration both atomic-level information and the contributions from functional groups or subgraph-level structural information. Distinguishing itself from traditional GNNs, the GSBF model seems to treat the molecular graph as an assembly of substructure graphs obtained via a decomposition process. Each subgraph is then represented using an embedding layer, providing an vector-embedding representation. Intriguingly, the authors combine information from these two branches (atom-level & subgraph-level branches) using a soft-attention mechanism for the final prediction. The paper also introduced a self-supervised pre-training strategy by masked prediction for the subgraph tokens.

**Strengths:**

While many studies have explored how to incorporate the compositional and hierarchical nature of molecules into Graph Neural Networks (GNNs), it's evident that embedding such inductive biases into model design can enhance accuracy in molecular property predictions. In this light, the paper's approach of explicitly considering substructural vocabulary appears both significant and promising. Notably, the use of structural decomposition algorithms like principle subgraph (Kong et al, 2022) and BRICS (Degen et al, 2008) suggests that they might serve as a more effective inductive bias than merely considering all subgraphs indiscriminately.

**Weaknesses:**

The idea of utilizing both atom-level and subgraph-level branches in GNN design has already been explored in existing research, and this paper falls short in providing comprehensive comparisons with these existing studies. For effective molecular generation, incorporating substructural compositional information can almost be seen as imperative. For instance, in the widely recognized GNN study for molecules (Ref-1, Ref-2), the concept of decomposing graphs into junction trees and considering subgraphs in tree structures for generation was proposed early on. Such ideas have been simultaneously tested in molecular property predictions and are well-known in GNN studies focusing on molecules (Ref-3, Ref-4, and cited Zhang et al, NeurIPS2021). Additionally, in pre-training designs with molecular property prediction as a downstream task, Zhang et al already proposed the use of fragmentation-based decomposition such as BRICS and RECAP, which decompose molecular structures into subgraph motifs. These designs then also consider message passing on motif trees separately from the usual atom-level message passing as in the 'two branch (bilateral) approach.' Therefore, while the reported improvements in benchmark performance are intriguing, the superiority of this method remains unclear without a detailed comparison with these closely related techniques.

Ref-1: Jin et al, Junction Tree Variational Autoencoder for Molecular Graph Generation. (ICML2018)

Ref-2: Jin et al, Hierarchical Generation of Molecular Graphs using Structural Motifs (ICML2020)

Ref-3: Fey et al, Hierarchical Inter-Message Passing for Learning on MolecularGraphs. ICML2020 workshop on graph representation learning and beyond (GRL+)

Ref-4: You & Gao, Molecular representation learning via heterogeneous motif graph neural networks. (ICML2022)

**Questions:**

Q1. For the construction of the subgraph vocabulary, you've cited the principle subgraph extraction strategy (Kong et al., 2022) and BRICS (Degen et al., 2008). In this implementation, did you adopt the former?


Q2. The GSBF relies directly on this subgraph vocabulary in its architecture, so it has a dependency on the algorithm used for vocabulary generation. How does its performance change when compared with, for example, BRICS?

Q3. Zhang et al (2021) is cited, and I felt there are many similarities, such as the use of motif vocabulary (using BRICS) and message-passing on the motif tree, in addition to standard atom-level message passing. In fact, the proposed method seems to be more naive than Zhang's. I'm unclear as to why the proposed method outperforms. Can you provide additional information on the main design differences and why these differences led to performance improvements over existing methods?

Q4. Other than Zhang's, there are also existing studies based on bilateral message passing of atom-level and subgraph-level representation. For example, have you compared with those methods like Ref-3 and Ref-4 above? If there are unique features in this paper that bring about superiority in prediction performance, please clarify them.